# Characterization of Humanized Mouse Model of Organophosphate Poisoning and Detection of Countermeasures via MALDI-MSI

**DOI:** 10.3390/ijms25115624

**Published:** 2024-05-22

**Authors:** Caitlin M. Tressler, Benjamin Wadsworth, Samantha Carriero, Natalie Dillman, Rachel Crawford, Tae-Hun Hahm, Kristine Glunde, C. Linn Cadieux

**Affiliations:** 1The Johns Hopkins University Applied Imaging Mass Spectrometry Core and Service Center, Division of Cancer Imaging Research, The Russell H. Morgan Department of Radiology and Radiological Science, The Johns Hopkins University School of Medicine, Baltimore, MD 21205, USA; 2United States Army Medical Research Institute of Chemical Defense, Aberdeen Proving Ground, Gunpowder, MD 21010, USA; 3Oak Ridge Institute for Science and Education, Oak Ridge, TN 37830, USA; 4The Sidney Kimmel Comprehensive Cancer Center, The Johns Hopkins University School of Medicine, Baltimore, MD 21205, USA; 5Department of Biological Chemistry, The Johns Hopkins University School of Medicine, Baltimore, MD 21205, USA

**Keywords:** MALDI imaging, organophosphate, reactivator, 2-PAM, atropine

## Abstract

Organophosphoate (OP) chemicals are known to inhibit the enzyme acetylcholinesterase (AChE). Studying OP poisoning is difficult because common small animal research models have serum carboxylesterase, which contributes to animals’ resistance to OP poisoning. Historically, guinea pigs have been used for this research; however, a novel genetically modified mouse strain (KIKO) was developed with nonfunctional serum carboxylase (Es1 KO) and an altered acetylcholinesterase (AChE) gene, which expresses the amino acid sequence of the human form of the same protein (AChE KI). KIKO mice were injected with 1xLD_50_ of an OP nerve agent or vehicle control with or without atropine. After one to three minutes, animals were injected with 35 mg/kg of the currently fielded Reactivator countermeasure for OP poisoning. Postmortem brains were imaged on a Bruker RapifleX ToF/ToF instrument. Data confirmed the presence of increased acetylcholine in OP-exposed animals, regardless of treatment or atropine status. More interestingly, we detected a small amount of Reactivator within the brain of both exposed and unexposed animals; it is currently debated if reactivators can cross the blood–brain barrier. Further, we were able to simultaneously image acetylcholine, the primary affected neurotransmitter, as well as determine the location of both Reactivator and acetylcholine in the brain. This study, which utilized sensitive MALDI-MSI methods, characterized KIKO mice as a functional model for OP countermeasure development.

## 1. Introduction

Organophosphorus (OP) poisoning through warfare, terrorism, and accidental exposure can result in death due to the inhibition of acetylcholinesterase (AChE) at the active site [1]. The OP agent binds to AChE through the active site serine, which blocks the active site, preventing the breakdown of acetylcholine (ACh) into choline [1]. The build-up of acetylcholine leads to a cholinergic crisis, resulting in difficulty in breathing, seizures, and death [1,2,3]. This process can be reversed with the use of oxime reactivators, which scavenge the bound OP, releasing the OP from the active site [4,5]. Currently, pralidoxime, also known as 2-PAM, and atropine are the fielded countermeasures for OP intoxication, where 2-PAM is the oxime-based Reactivator and atropine is the alkaloid-based muscarinic antagonist [6]. To date, despite there being a well-understood mechanism of action, many questions remain surrounding the treatment of OP intoxication, including if oxime-based reactivators can cross the blood–brain barrier.

Previously, guinea pigs were generally used as the preferred small animal model for OP intoxication experiments; however, a humanized mouse strain has been developed that incorporates two specific genetic modifications combined on a defined genetic background (C57BL/6J) [7]. Specifically, a mouse strain (C57BL/6-*Ces1c^tm1.1Loc^*/J, referred to as Es1 KO mice) in which the gene-expressing serum carboxylesterase was interrupted such that mature protein was not expressed [8] was cross-bred with a mouse strain (C57BL/6-*AChE^tm1.1Loc^*/J, referred to as AChE KI mice) in which the gene encoding acetylcholinesterase (AChE) was altered to express the amino acid sequence of the human form of the same protein. The resulting mouse strain (AChE KI/Es1 KO, or KIKO) incorporated both of these modifications into a single animal model that addresses two major concerns for the chemical warfare agent research community [7].

The lack of functional serum carboxylesterase (CaE) in these mice mirrors humans and nonhuman primates, both of which do not express CaE. Most other commonly used small animal research models (other strains of mice, rats, and guinea pigs) express serum CaE, which is known to directly contribute to those animals’ resistance to toxicity from OP compounds [9,10,11,12]. Additionally, the presence of serum CaE in mice has been shown to affect the pharmacokinetic profile and efficacy of other pharmaceuticals [13].The production of AChE with the same amino acid sequence as that of the human form of the enzyme in KIKO mice presents a unique opportunity for this animal model to study compounds that interact directly with AChE. Previous research has revealed that, although AChE performs the same function in all animals, minor amino acid differences across species cause the enzyme to react quite differently to small molecules intended to restore the native activity of the OP-inhibited enzyme, and inhibition and aging are the main causes of nerve agent intoxication [14,15,16,17,18,19,20].

With the advent of this novel mouse model, we sought to use matrix-assisted laser desorption ionization (MALDI) mass spectrometry imaging (MSI), which is a powerful, label-free molecular imaging approach combining the power of mass spectrometry with spatial resolution [21]. MALDI imaging uses the molecular mass, which is an inherent molecular parameter, to determine and spatially localize a large range of different biomolecules in tissue sections, organoids, and cells. Similar to widely used histology and immunohistochemistry approaches, MALDI imaging utilizes tissue sections or cells that are placed on microscopy slides for imaging experiments. The tissue is then raster-scanned with a laser pixel by pixel, and a mass spectrum with hundreds of peaks is acquired at each pixel [22]. These spectra can then be displayed as ten-to-hundred-of-ion density maps, which are basically tissue images of each individual ionized molecule from the tissue detected in the mass spectral domain, visualizing the spatial distribution of these molecular ions as MALDI images [23]. MALDI MSI is a versatile technology that can be used to measure metabolites [23,24,25], neurotransmitters [26], drugs [24,27], N-glycans [28,29], lipids [23,30,31], peptides [24,28], and proteins [24]. MALDI MSI is becoming increasingly more commonplace; however, it is largely absent from the Chem-Bio Defense community. While some studies in pesticide research have been published, the organophosphorus nerve agent (OPNA) field has been largely left untouched. This is particularly surprising considering that many early experiments on reactivators and OPNA exposure were performed using MALDI time of flight (ToF) experiments. We aim to apply this very powerful technique to this field to study the effects and treatment of OPNA exposure.

## 2. Results

In order to determine the potential of the KIKO model with MALDI MSI, we designed an experiment to measure the presence of the currently fielded Reactivator [6], 2-PAM, and acetylcholine in animals that had been exposed to an OPNA that has been used in recent attacks including the Tokyo subway attack in 1997 [32] and in the Syrian civil war as recently as the early 2000s [32]. Briefly, animals were injected with OPNA or the Vehicle control. After one to three minutes, animals were injected with either Reactivator or the Vehicle control (Figure 1). Animals were then humanely euthanized at fifteen minutes post-Reactivator injection or at the time of death (Figure 1). Brains and hearts were harvested for MALDI MSI experiments.

As part of the initial imaging experiment in which atropine was not used, we were able to observe low levels of Reactivator at *m*/*z* 137.1 in both the Vehicle/Reactivator and OPNA/Reactivator experimental groups but not in the Vehicle/Vehicle or OPNA/Vehicle groups (Figure 2A). These data indicate that oxime Reactivator is able to cross the blood–brain barrier, despite containing a permanent positively charged moiety. We were able to confirm the presence of Reactivator in the brain tissue using MSMS as compared with a standard collected from a target plate experiment (Appendix A). The signal intensity based on the peak area was plotted and the signal was increased in both the Vehicle/Reactivator and OPNA/Reactivator experimental groups (Figure 3A). These data, when separated by sex, show more Reactivator in female mice than male mice (Figure 3B). To investigate if this disparity was a result of early death, the time of death was plotted, and no correlation was observed between the presence of Reactivator in the brain and time to death (Figure 3C). This was particularly notable in the Vehicle/Reactivator experimental group where all six animals lived to the 15 min timepoint; however, the female mice had more Reactivator present in the brain tissue than male mice (Figure 3B,C). We also observed that male mice generally did not survive up to the 15 min time point in the OPNA/Reactivator group and had less Reactivator than the female mice (Figure 3B,C).

As expected, the presence of OPNA increased the acetylcholine level (*m*/*z* 146.1, Appendix A) in exposed animals as compared with that in the Vehicle control (Figure 2B and Figure 3B). We observed a difference in acetylcholine levels between male and female animals, with male animals generally having more acetylcholine than female animals (Figure 3B). We did, however, observe an inverse correlation with acetylcholine levels and time to death, unlike the case for the Reactivator concentration (Figure 3C). Although the observed trends were not found to be statistically significant, these data indicate that acetylcholine levels may be either sex-dependent or dependent on the time to death. 

To further investigate this question, we repeated the experiment, this time with the presence of 1.35 mg/kg of atropine to increase the number of animals surviving to the 15 min time point. We again observed the presence of Reactivator in the brains of the two groups that received Reactivator, regardless of the presence of OPNA (Figure 4A). Reactivator was present in two primary hotspots, one in the cerebellum, and one in the forebrain (Figure 4A). As with the previous experiment, acetylcholine was increased in groups that received OPNA (Figure 4B). 

In groups that received Reactivator, we found a significant increase in *m*/*z* 137.1, which agrees with the expectations for Reactivator as determined via MSMS (Figure 5A and Appendix A). While there was a low level of the endogenous signal at 137.1, we were able to see a clear, statistically significant increase associated with the treatment with Reactivator. In this experiment, no statistically significant difference between male and female mice was seen in the groups that received Reactivator (Figure 5A). We also observed a statistically significant difference in acetylcholine in animals that received OPNA as compared with those that did not (Figure 5B); however, we did not observe a difference between male and female mice in terms of the amount of acetylcholine. These data indicate the differences observed in the previous experiment may have been, in part, due to the differences in the time of death between male and female animals. In this data set, we do not observe a correlation of acetylcholine with time of death and the only animals to not reach the fifteen minute time point were animals that only received atropine (Figure 5C). 

In the OPNA/Reactivator group, we saw localization of the Reactivator across the brain (Figure 6). Reactivator was generally distributed across the brain (Figure 6B and Appendix A) but hot spots of Reactivator were visible. When aligned with an H&E stain of the same section (Figure 6A and Appendix A), the hot spots aligned with the fourth ventricle, the third ventricle, and the hypothalamus. Both the third and fourth ventricles are involved in the production and movement of cerebrospinal fluid (CSF). Reactivator was also found in the cerebellum, the fornix, and the hypothalamus. The ACh signal was primarily located in the hippocampus of the brain (Figure 6).

In order to determine the relative amount of Reactivator in the brain, a tissue mimetic model was developed for quantitative MALDI imaging (Figure 7). The tissue mimetic model is an established method for determining the absolute quantification of drugs via MALDI imaging [33,34]. The tissue mimetic model was generated from KIKO mouse liver tissue with a function of y = 2024x + 43.592 and a coefficient of determination of R^2^ = 0.893 using weighted linear regression (Figure 7B). The limit of detection was calculated as 0.0259 µg/mg of tissue. In the case of brain tissue, the detection of Reactivator was below the limit of quantification for this tissue mimetic model (Figure 7A). However, as a proof of principle, we also measured Reactivator in the heart tissue from the same animals. Reactivator in the heart tissue was calculated to be (left) 0.076 µg/mg of tissue and (right) 0.092 µg/mg of tissue (Figure 7A). These data demonstrate that it is possible to quantify Reactivator in the system; however, the level detected in the brain tissue was too low to suggest reliable quantification. 

Finally, to demonstrate that Reactivator detected in the brain imaging experiment was present in the brain tissue versus any residual blood, we performed CD31 staining for blood vessel endothelial cells (Figure 8). We performed staining in both male and female brain tissue. Results of this staining indicate that while some overlap in signals was present, hot spots of Reactivator material did not correspond with any major signal from CD31 staining. Further, we examined the dataset for the presence of heme, which was detected at *m*/*z* 616.2 (Appendix A). As a primary component of blood, we would expect to see an overlapping signal between Reactivator and heme if Reactivator was primarily found in the blood, but this was not the case.

## 3. Discussion

These experiments are the first of their kind, utilizing both a novel mouse model for OPNA poisoning as well as MALDI MSI to determine the localization of Reactivator and ACh in the brain. Although the currently fielded Reactivator, 2-PAM, was used in this study, the experiments described here could be used to evaluate any Reactivator/OPNA combination in the future. While it has long been suspected that 2-PAM may enter the brain after OPNA exposure and treatment, this is the first example of a direct, unlabeled measurement of 2-PAM within the brain. These data suggest that 2-PAM, a permanently positively charged oxime molecule, is able to cross the blood–brain barrier, regardless of exposure to OPNA. These data are counter-intuitive to the prevailing consensus that 2-PAM should not be able to cross the blood–brain barrier [35]. However, our data clearly show that even without the presence of OPNA, 2-PAM is still able to enter the brain, demonstrating it clearly crosses the blood–brain barrier. 

Reactivator 2-PAM was found throughout the brain with hot spots in locations that also contained CSF such as the third and fourth ventricle. Previous reports have shown that 2-PAM enters the brain at a rate of approximately 10%, but these studies were conducted using traditional mass spectrometry, which does not preserve spatial resolution [36]. These data show brain penetrance throughout the brain and CSF, allowing for potential reactivation throughout the central nervous system. 

The methods used in this study also allow for the simultaneous, direct measurement of ACh, the metabolite primarily affected by OPNA exposure in a spatially resolved manner. Build-up of OPNA in the brain leads to an inhibition of AChE, which breaks down ACh. As expected, we observe an increase in ACh to be associated with OPNA exposure. We primarily observed ACh throughout the brain; however, a hot spot was observed, the hippocampus, which has previously been reported [37]. The presence of Reactivator did not statistically significantly reduce the amount of ACh present in the brain at the fifteen minute timepoint; however, we did observe a small decrease, which may indicate that Reactivator was beginning to reactivate AChE. This observation is consistent with previously observed brain penetrance studies that show that some 2-PAM, but not enough for therapeutic benefits, enters the brain [36]. Further studies will contain longer timepoints that will more deeply explore the timeline of reactivation in vivo. 

We were able to determine the relative quantitation for 2-PAM in the brain using the previously established tissue mimetic model system [33,34]. Tissue mimetic models are the current standard for the absolute quantification of drugs via MALDI imaging. While 2-PAM, specifically, has not been previously measured, a number of different drugs for different applications have [33,34]. We found the amount of 2-PAM to be lower than the limit of quantification, meaning there was less than 0.0259 µg 2-PAM/mg of tissue in the brain of both male and female animals. This is likely not a high enough concentration of 2-PAM to effectively reactivate AChE in a timely manner, even in localized “hot spots” of higher concentrations of the countermeasure.

Finally, we demonstrated that a small amount of 2-PAM does leave the bloodstream and enter the brain tissue. This conclusion was reached using a combination of immunofluorescence and the endogenous heme signal present in the brain tissue sections. CD-31 staining was used for staining blood vessels in tissue, including brain and tumor [38,39]. The 2-PAM signal did not overlap with either the CD-31 endothelial marker or the endogenous heme signal. These data support the idea that 2-PAM is present in the brain tissue rather than residual blood present in the brain.

A significant amount of research is currently being conducted to develop novel oxime-based reactivators that more readily cross the BBB [40]. These molecules include 2-PAM prodrugs [41], novel delivery methods [42,43], and novel oxime molecules [40,44]. This work demonstrates a pipeline that can be used to evaluate brain penetrance including absolute quantification and the localization of novel oxime countermeasures with or without OPNA exposure. The information this pipeline provides is particularly crucial considering that these reactivators cannot be tested on humans with OPNA exposure until fielded [44].

These experiments demonstrate the powerful combination of the KIKO mouse model and MALDI mass spectrometry imaging in identifying and characterizing known and novel reactivators of OPNA-inhibited AChE. To date, pyridinium oximes are the primary chemical moiety employed as reactivators that are ideal for MALDI MSI analysis [35] due to the permanent positive charge, meaning that the molecule is already ionized. Future work in this area will entail examining other known oxime-based reactivators and novel reactivators that show promise in vitro. Further studies should include examining other molecular classes to determine the side effects of OPNA exposure and treatment.

## 4. Materials and Methods

Materials—All materials and reagents were purchased from Sigma-Aldrich (St. Louis, MO, USA) unless otherwise noted. All reagents were used without further purification. All solvents used were HPLC-grade or higher. For all experiments described in this study, “Vehicle” is water (Hospira, Inc., Lake Forest, IL, USA), “OPNA” is sarin (obtained from the U.S. Army Combat Capabilities Development Command (DEVCOM) Chemical Biological Center, Edgewood, MD USA), and “Reactivator” is 2-PAM (Baxter Healthcare Corp., Bloomington, IN, USA.

Animal experiments—The animal study protocol was approved by the Animal Care and Use Committee at the United States Army Medical Research Institute of Chemical Defense. Male and female adult KIKO mice (14 to 16 weeks) were assigned to one of the following groups (n = 3 males and n = 3 females for a total of n = 6 animals per group): (1) Negative Control (Vehicle/Vehicle), (2) Treatment Only (Vehicle/Reactivator), (3) Exposure Only (OPNA/Vehicle), or (4) Exposure and Treatment (OPNA/Reactivator). Animals were randomly sorted into experimental groups. Each animal received two subcutaneous injections with the first injections consisting of a median lethal dose of OPNA or a similar volume of vehicle administered in the scruff. The second injection, administered one to three minutes later in the flank, consisted of Reactivator (35 mg/kg) or a similar volume of Vehicle. Animals were then euthanized 15 min post-Reactivator injection or prior to succumbing to death (early endpoint) via isoflurane and decapitation. The brain and heart tissues were immediately collected and cryopreserved in liquid nitrogen for 2 to 3 min. Brains were not perfused prior to freezing. Samples were then kept on dry ice until placed in long-term storage at −80 °C where they were maintained until processing for imaging. 

During the initial exposure experiment, it was found that a significant number of mice administered OP suffered early-end-point deaths, confounding the comparative analysis of the brain penetrance of Reactivator. To address this issue, the exposure experiment was modified to include an allometrically scaled human-equivalent dose of atropine (1.35 mg/kg). The atropine was injected subcutaneously in the opposite flank as Reactivator immediately following Reactivator dosing. Tissue was again collected and cryopreserved 15 min after Reactivator administration or at an early end point as described previously.

Tissue Mimetic Model—A tissue mimetic model was generated as previously described [33,34]. The tissue mimetic model was generated from KIKO mouse livers that were harvested and preserved with a freeze clamp. Livers were crushed into small pieces on dry ice and 500 milligrams of liver was weighed into 2 milliliters of Precellys Lysing kit hard tissue homogenizing CK28 tubes. Tissue was homogenized on a Precellys Evolution bead homogenizer with the cryolysis attachment. The system was cooled using dry ice per the manufacturer’s recommendation. Tissue was homogenized using the preset elastic tissue cycle (speed: 6800 rpm, cycle: 4 × 30 s, pause; 45 s). Tubes were centrifuged at 15,000 rpm for one minute. Reactivator standards (10 µL) were added to each tube, and the tissue was homogenized using the same cycle. Tubes were centrifuged at 15000 rpm for one minute before transferring each standard homogenate to a biopsy mold. Homogenates were frozen overnight at −80 °C before cryosectioning. 

Cryosectioning—Indium tin oxide (ITO) slides (Delta Technologies, Loveland, CO, USA) were washed with hexanes (10 min with sonication) and ethanol (10 min with sonication). Brains, the tissue mimetic model, and ITO slides were equilibrated to −20 °C for cryosectioning. Brains were mounted with Shandon M1 embedding media (Fisher Scientific, Waltham, MA, USA) and placed facing the midline for sectioning. Ten-micron-thick sections were taken starting at the midline and thaw-mounted onto cold ITO slides. 

Matrix Application—The matrix was applied using an HTX M3+ sprayer (HTX Technologies, Chapel Hill, NC, USA). 2′,4′,6′-Trihydroxyacetophenone monohydrate (THAP) was used as a matrix at a 10 mg/mL concentration in 70% acetonitrile with 0.1% trifluoroacetic acid (TFA) [45]. The sprayer method is as follows: nozzle temperature: 50 °C; pressure: 10 psi; flow rate: 120 µL/min; track spacing: 2.5 mm; 6 passes; VV pattern; a 2 s drying time. This resulted in a matrix density of 2.4 × 10^−3^ mg/mm^3^ and a linear flow rate of 1 × 10^−4^ mL/mm. 

MALDI Imaging—Samples were measured on Bruker Rapiflex ToF/ToF (Bruker Daltonics, Billerica, MA, USA) in reflection positive mode from *m*/*z* 100 to 1000 with 200 laser shots per pixel using a 50-micron-laser single laser (50-micron spot size; 50-micron step size). 

Tandem MS—Tandem MS was measured on Bruker RapifleX ToF/ToF (Bruker Daltonics, Billerica, MA, USA) in MS/MS mode using a single-laser configuration with a resultant field size of 50 × 50 microns. All fragmentation spectra were collected with collision-induced decay (CID) turned on and argon as a collision gas. The Reactivator standard was dissolved in water and spotted (1 µL) onto a ground steel target plate with THAP as a matrix, as noted above (1 µL). On tissue, MSMS was performed on the same section that was imaged in an area with a high concentration of the peak of interest. All MSMS experiments used a filter window of ±1 Da. MSMS spectra can be found in the Appendix A.

Data Analysis—All imaging data sets were background-corrected in FlexImaging (v5.0, Bruker Daltonics). All imaging experiments were loaded into SCiLS Lab (v2023b Pro, Bruker Daltonics) using the default settings with the parameters from FlexImaging. The data were normalized to the total ion count (TIC), and a peaklist was generated and aligned in SCiLs lab. 

Immunofluorescence Staining– Immunofluorescence staining was adapted from previous studies [38,39]. After MALDI imaging, brain sections were washed in 100% ethanol for at least 24 h to remove the matrix and were subjected to immunostaining. Initially, the sections were rinsed with ethanol and subsequently fixed in freshly prepared 4% paraformaldehyde (PFA) solution at room temperature for 15 min. Following fixation, the brain slices underwent incubation in a solution containing 10 mM citrate (pH 6.0) and 0.01 M PBS (pH 7.2). This was followed by an incubation step with a mixture consisting of 20% goat serum, 1% BSA, 0.3 M glycine, and 0.01 M PBS (pH 7.2) for 24 h at 4 °C. The brain slices were then exposed to a primary antibody targeting CD31 (R&D systems, Minneapolis, MN) for 24 h at 4 °C.

For CD31 immunostaining, subsequent incubation was carried out using Goat anti-Mouse IgG (H + L) Highly Cross-Adsorbed Secondary Antibody, Alexa Fluor™ 647 (1:200, Invitrogen, Carlsbad, CA), at 37 °C for an additional 3 h. Following this, the samples were stained with DAPI for 2 min and washed with PBS prior to mounting.

Immunofluorescence Microscopy—Mouse brain vasculature was visualized using an Olympus Slideview VS200 slide scanner and Olyvia 3.3 software equipped with a 40× objective (Johns Hopkins Applied Imaging Mass Spectrometry (AIMS) Core facility). Additionally, 10 μm thick mouse brain slices stained with DAPI and CD31 were imaged using a 40× objective under excitation wavelengths of 377 nm and 647 nm.

## 5. Conclusions

In conclusion, we propose the KIKO mouse model in combination with MALDI MSI as a model to directly measure both reactivators, such as 2-PAM, and the level of acetylcholine in the brain of OPNA-exposed animals via MALDI MSI. This is the first example of an imaging-based approach being used to measure 2-PAM in the brain directly without the use of a label. This pipeline can be used to examine oxime reactivators, including currently fielded as well as novel ones, to determine relative amounts and localization within the brain.

## Figures and Tables

**Figure 1 ijms-25-05624-f001:**
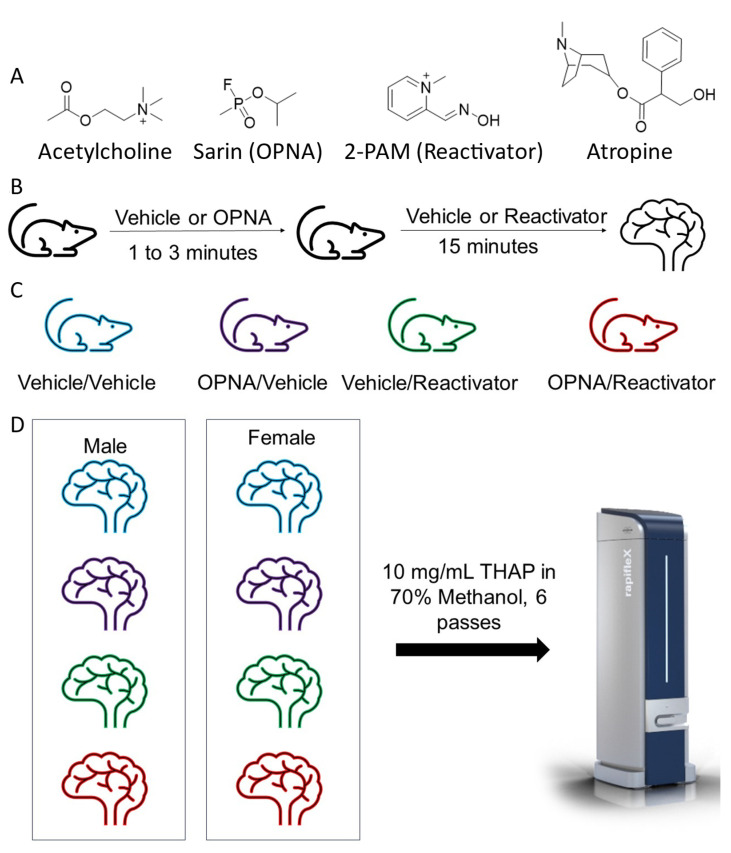
Experimental design. (**A**) Compounds used in experiments. (**B**) Diagram showing that animals were treated with vehicle or OPNA (sarin) and that after 1 to 3 min they were treated with Vehicle or Reactivator. After 15 min, animals were euthanized, and brains and hearts were removed. (**C**) The experimental groups in this study. (**D**) The MALDI MSI slide layout with one brain from each group per slide. Slides contained either all male or all female animals and were coated with 10 mg/mL of THAP in 70% methanol prior to imaging on Bruker RapifleX.

**Figure 2 ijms-25-05624-f002:**
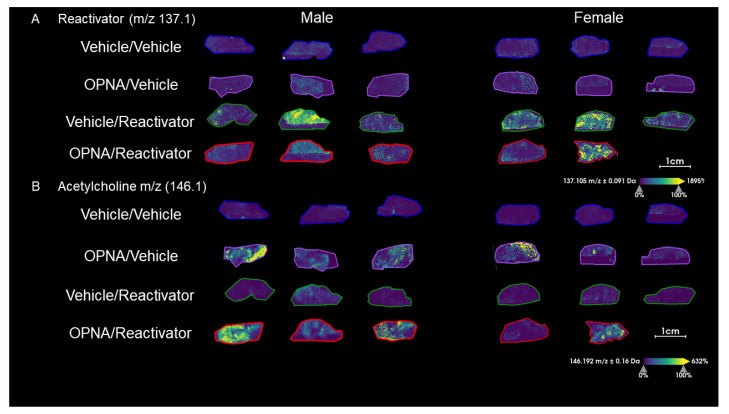
(**A**) Images of Reactivator (*m*/*z* 137.1) in male (left) and female (right) KIKO mouse brains without atropine. (**B**) ACh (*m*/*z* 146.1) in male (left) and female (right) KIKO mouse brains without atropine.

**Figure 3 ijms-25-05624-f003:**
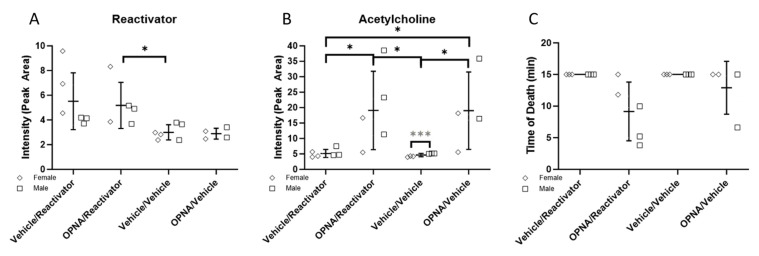
(**A**) Intensity (peak area of *m*/*z* 137.1) of Reactivator plotted by both experimental group and sex. (**B**) Intensity (peak area of *m*/*z* 146.1) of ACh plotted by experimental group and sex. (multiple *t*-test; * *p* < 0.05, *** *p* = 0.0009). (**C**) Time of death plotted by experimental group.

**Figure 4 ijms-25-05624-f004:**
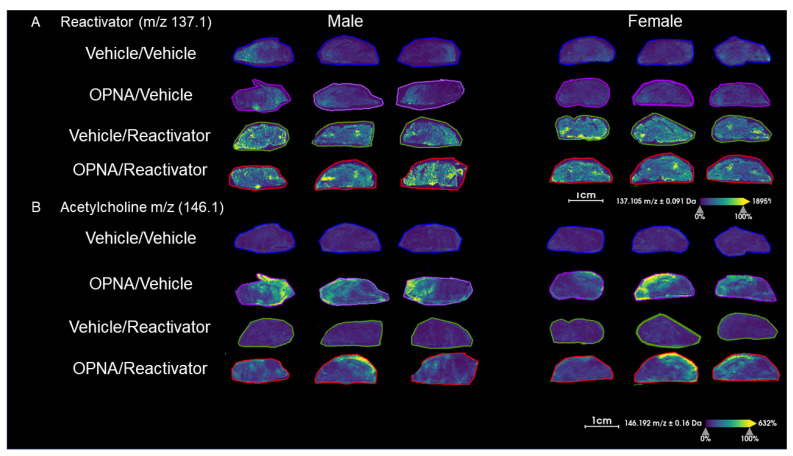
(**A**) Images of Reactivator (*m*/*z* 137.1) in male (left) and female (right) KIKO mouse brains with atropine. (**B**) ACh (*m*/*z* 146.1) in male (left) and female (right) KIKO mouse brains with atropine.

**Figure 5 ijms-25-05624-f005:**
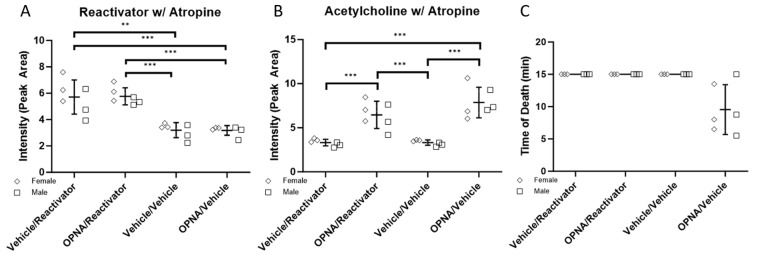
(**A**) Intensity (peak area of *m*/*z* 137.1) of Reactivator with atropine plotted by both experimental group and sex. (**B**) Intensity (peak area of *m*/*z* 146.1) of ACh with atropine plotted by experimental group and sex. (Multiple *t*-test; ** *p* = 0.001, *** *p* ≤ 0.0007). (**C**) Time of death plotted by experimental group with atropine.

**Figure 6 ijms-25-05624-f006:**
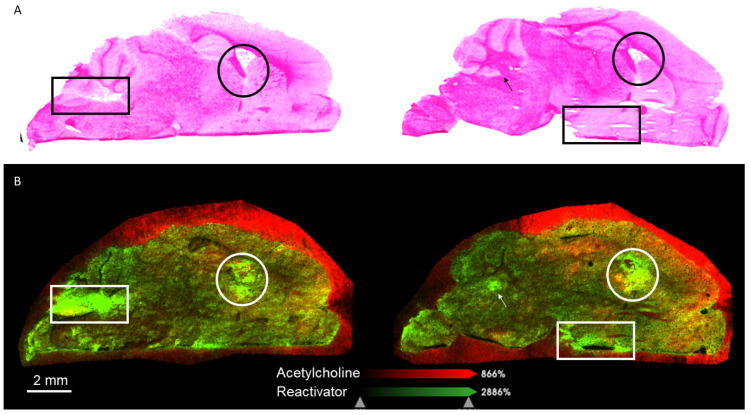
(**A**) Sagittal H&E stains of male (left) and female (right) brains. (**B**) Overlays of ACh (red, *m*/*z* 146.1) and Reactivator (green, *m*/*z* 137.1) of the fifteen minute timepoint in OPNA and Reactivator-treated brains with atropine. The square, circle, and arrow indicate areas of high Reactivator content.

**Figure 7 ijms-25-05624-f007:**
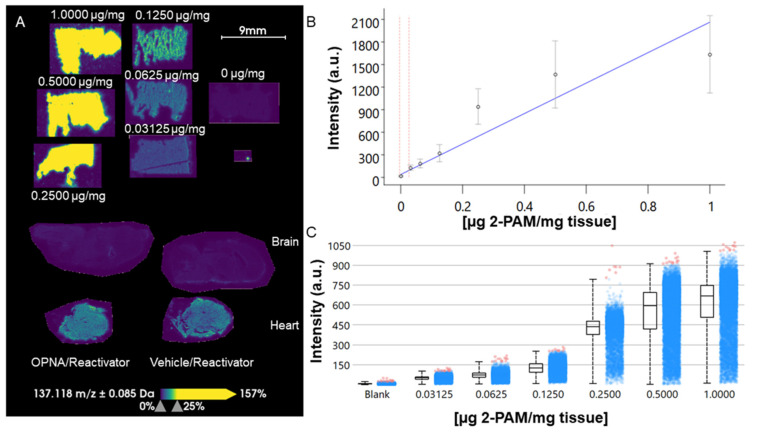
(**A**) Tissue mimetic model, brains, and hearts from groups that received Reactivator. (**B**) Standard curve generated from tissue mimetic model with the red lines showing the limit of detection (0.0259 µg 2-PAM/mg of tissue) and the limit of the blank (0 µg 2-PAM/mg of tissue). (**C**) Intensity box plot of tissue mimetic model.

**Figure 8 ijms-25-05624-f008:**
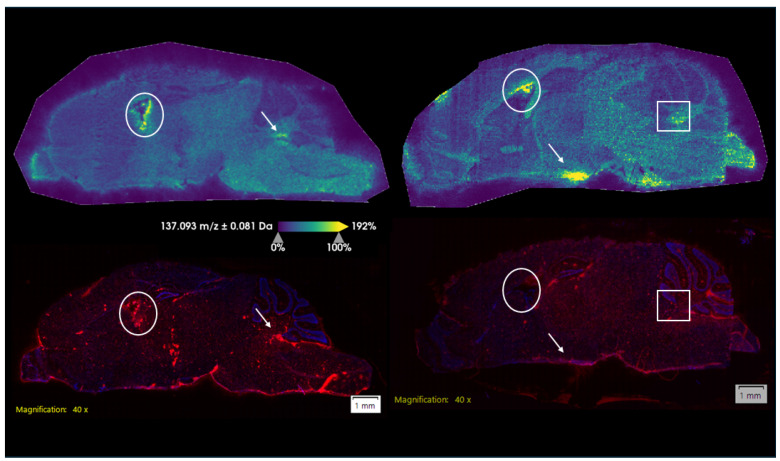
Female (left) and male (right) brain sections showing 2-PAM (top, *m*/*z* 137.1) and CD31 immunofluorescence staining (bottom) on the same brain section. The circle, square, and arrow mark areas of high Reactivator signal.

## Data Availability

The data presented in this study are available on request from the corresponding author.

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
