# Peer review of "Characterization of Humanized Mouse Model of Organophosphate Poisoning and Detection of Countermeasures via MALDI-MSI"

_ijms, 2024, doi:10.3390/ijms25115624_

Round 1

Reviewer 1 Report

Comments and Suggestions for Authors

Dear Authors, dear Editor,

Draft “ijms-2981401-peer-review-v1 Characterization of a Humanized Mouse Model …” reports on MALDI-imaging studies performed in the brain and heart of trans-genic mice featured to be closely similar to the human with respect to response to organophosphorus anti-cholinesterase neurotoxic agents, such as the warfare nerve gases and an important class of insecticides. Mice were first intoxicated with one “nerve gas” (with or without concurrent administration of atropine as prophylactic antidote), then treated with a standard post-exposure antidote, pralidoxime, and brain and heart slices of the euthanized animals were measured by MALDI-imaging to localize and estimate levels of the administered compounds. Analyses aim at highlighting localization and levels of acetyl-choline (the neurotransmitter m/z 146.1) and pralidoxime (the standard post-exposure antidote m/z 137.1) that showed differences between male and female animals. Due to the very small number of tested animals and to the well-known low quantitative reliability of MALDI-imaging, results are, at best, proof-of-principle and suggestive.

In principle, the work is interesting and worth publishing in Toxics; however, there are points where it should be made clearer. There are some marginal mistakes, that nevertheless do not obscure meaning.

It would be better to make a figure and scheme of the main substances involved: atropine, acetylcholine, pralidoxime (if this is the “Reactivator”, this should be stated clearly; if it is another substance, too, unless it is stated that the structure cannot be disclosed). The same is for OPNA: if it is sarin, use that name only.

The spectra reported in the supplementary are very untidy and, in the case of m/z 87, the ion structure is not correct: at least, it should be in the 5-membered cyclic 1,3-dioxo form. All reported mass spectra should be annotated to highlight structure-distinctive fragments.

Figures 2, 4 and 3,5 which make the core of the work, should carry correct compound labels. In Figures 3,5 it is not useful doubling graphs, because those with separately labelled male and female animals convey all the information. Use the saved space to make them larger. Can you see the signal of atropine in any of the imaging spectra? If so, you may add them to Figure 4. As for the meaning of your observation, is there any other evidence of higher levels of Ach in the female? BTW, I wonder why you did not include a control-no treatment MALDI-im KIKO group in the study.

As for quantification in MALDI-imaging, it is very fair of you to show the “calibration line” in Figure 7. If you ever used it for estimating PAM levels, it would be useful to understand “where” they are located in the calibration line, because uncertainty is huge. As for >Figure 8, where is the plot of the heme signal? Of course, your observation that PAM crosses the BBB with evidence from MALDI-im is very important and worth for presenting better evidence.

As you understand, only minor revision and some typo check is needed to improve this draft.

Kind regards

Comments on the Quality of English Language

just some typo check

Author Response

Reviewer 1

It would be better to make a figure and scheme of the main substances involved: atropine, acetylcholine, pralidoxime (if this is the “Reactivator”, this should be stated clearly; if it is another substance, too, unless it is stated that the structure cannot be disclosed). The same is for OPNA: if it is sarin, use that name only.

            Response: Structures to figure 1 with naming language indicated in figure.

The spectra reported in the supplementary are very untidy and, in the case of m/z 87, the ion structure is not correct: at least, it should be in the 5-membered cyclic 1,3-dioxo form. All reported mass spectra should be annotated to highlight structure-distinctive fragments.

Response: Unfortunately, the nature of MSMS from tissue is that the spectra are often untidy. We have updated the figures with the necessary fragments and updated the structure for m/z 87.

Figures 2, 4 and 3,5 which make the core of the work, should carry correct compound labels. In Figures 3,5 it is not useful doubling graphs, because those with separately labelled male and female animals convey all the information. Use the saved space to make them larger. Can you see the signal of atropine in any of the imaging spectra? If so, you may add them to Figure 4. As for the meaning of your observation, is there any other evidence of higher levels of Ach in the female? BTW, I wonder why you did not include a control-no treatment MALDI-im KIKO group in the study.

Response: The labelling of figures was done to correspond to the language used throughout the manuscript text which is done for two reasons: 1) language guidance was provided by the government laboratory that participated in this effort and 2) the use of the language better illustrates that this method of evaluation could be applied universally to any OPNA or Reactivator. For clarity the following statement has been added to the Methods section: “For all experiments described in this study, “Vehicle” is water, “OPNA” is sarin, and “Reactivator” is 2-PAM.”

Figures 3 and 5 have been updated to combine the male/female and overall data into single graphs.

Other than the single observation noted in Figure 3, no additional differences between the Ach levels of males and females were noted.

The group labelled “Vehicle/Vehicle” in all figures constitutes a control (no treatment, no OPNA) group for the study. An additional control group with no treatment but receiving OPNA (“OPNA/Vehicle”) was included in the study as well.

As for quantification in MALDI-imaging, it is very fair of you to show the “calibration line” in Figure 7. If you ever used it for estimating PAM levels, it would be useful to understand “where” they are located in the calibration line, because uncertainty is huge. As for >Figure 8, where is the plot of the heme signal? Of course, your observation that PAM crosses the BBB with evidence from MALDI-im is very important and worth for presenting better evidence.

Response: We are not estimating the concentration of the 2-PAM as it is lower than the limit of quantification. The figure with the heme can be found in the supporting information.  

Reviewer 2 Report

Comments and Suggestions for Authors

The paper is quite interesting, however some issues should be corrected in order to improve the manuscript and allow its further publication.

1. in the introduction, the Authors stated about antidotum for OP- induced toxidrome. In fact oximes are found effective, however, the first treatment is based on the use of atropine. Therefore, this sentence should be corrected a bit.

2. Please provide the approval number given by the Ethical Committee, as the Authors used animals 

3. Please state whether animals were randomly divided into experimental groups. 

4. Since the Authors faced some problems with animals, as they suffered much, how many animals were finally taken for the study?

5. Please specify what was the vehicle used in the study: saline or water? In the methodology section, the Authors stated that saline was used, while in the results water is presented

6. Legend for Fig. 1 should have information on what is the vehicle, reactivate, etc. Also, what does A, B and C mean.

7. In line with the above-mentioned, in the methodology section the Authors stated that both female and male animals were used , and no information was given whether these animals were divided based on their sex. Therefore, please specify, whether the Authors studied brains (fig.1 c) of mixed groups or the results may reflect differences between animals in the aspect of their sex.

8. I'm quite confused with the results given in fig. 2. Again, the Authors stated that the number of animals per group was 3, so please explain the legend, or verify and correct the methodology section. Otherwise, is misleading

9. fig. 6 presents reactivator localization in the brain. however, the Authors should specify whether the image is lateral, etc

Comments on the Quality of English Language

minor changes are required

Author Response

Reviewer 2

  1. in the introduction, the Authors stated about antidotum for OP- induced toxidrome. In fact oximes are found effective, however, the first treatment is based on the use of atropine. Therefore, this sentence should be corrected a bit.

Response: We thank the reviewer for this information, however, while atropine is historically the first treatment, we are focused on the oxime for this paper rather than the atropine which is why the introduction reads the way it does.

  1. Please provide the approval number given by the Ethical Committee, as the Authors used animals 

Response: This has been added to the manuscript.

  1. Please state whether animals were randomly divided into experimental groups. 

Response: This has been added to the experimental section

  1. Since the Authors faced some problems with animals, as they suffered much, how many animals were finally taken for the study?

Response: The final number of animals 48 animals. We had 6 animals (3 male, 3 female) per group with four groups. The experiment was repeated a second time with atropine present. We did not replace animals that did not make it to the 15 minute time point.

  1. Please specify what was the vehicle used in the study: saline or water? In the methodology section, the Authors stated that saline was used, while in the results water is presented

Response: All sections have been updated to correctly indicate that the Vehicle is water.

  1. Legend for Fig. 1 should have information on what is the vehicle, reactivate, etc. Also, what does A, B and C mean.

Response: The figure 1 legend as been updated to read:

Figure 1. Experimental design. A. Compounds used in experiments. B. Animals were treated with vehicle or OPNA (sarin) and after 1 to 3 minutes they were treated with vehicle or reactivator. After 15 minutes, animals were euthanized and brains and hearts were removed. C. The experimental groups in this study. D. The MALDI MSI slide lay out with one brain from each group per slide. Slides contained either all male or all female animals and were coated with 10 mg/mL THAP in 70% methanol prior to imaging on a Bruker RapifleX.

  1. In line with the above-mentioned, in the methodology section the Authors stated that both female and male animals were used , and no information was given whether these animals were divided based on their sex. Therefore, please specify, whether the Authors studied brains (fig.1 c) of mixed groups or the results may reflect differences between animals in the aspect of their sex.

Response: Each group contained n = 3 males and n = 3 females leading to a combined n = 6 in each of the four experimental groups.

  1. I'm quite confused with the results given in fig. 2. Again, the Authors stated that the number of animals per group was 3, so please explain the legend, or verify and correct the methodology section. Otherwise, is misleading

Response: We have updated the methods section to read as the following:

Animal experiments – Male and female adult KIKO mice (14 to 16 weeks) were assigned to one of the following groups (n=3 males and n = 3 females for a total of n = 6 animals per group.) 1) Negative Control (Vehicle/Vehicle), 2) Treatment Only (Vehicle/Reactivator), 3) Exposure Only (OPNA/Vehicle), or 4) Exposure and Treatment (OPNA/Reactivator).

  1. fig. 6 presents reactivator localization in the brain. however, the Authors should specify whether the image is lateral, etc

Response: We have updated the figure legend to read:

Figure 6: A. Sagittal H&E stains of male (left) and female (right) brains. B. Overlays of ACh (red, m/z 146.1) and Reactivator (green, m/z 137.1) of fifteen-minute timepoint in OPNA and Reactivator treated brains with atropine. Square, circle, and area indicate areas of high Reactivator.

Reviewer 3 Report

Comments and Suggestions for Authors

The study entitled Characterization of a Humanized Mouse Model of Organophosphate Poisoning and Detection of Countermeasures via MALDI-MSI is a well-planned study with a reasonable contribution to the field. Humanization of Mouse Models in toxicological studies are not exactly new, nevertheless. I put forth the following suggestions.

1. The abstract needs to be refined and reported in past tense, considering that this is a report.  

2. For keywords, words already used in the title should not be reused as keywords.

3. The methodology needs to make reference to previous studies from which these procedures were obtained from.

3. The discussion needs to be done in comparison with previous studies. In the current state, it can not be considered a discussion of the results. For example:

Qiao, T., Xiong, Y., Feng, Y., Guo, W., Zhou, Y., Zhao, J., Jiang, T., Shi, C., & Han, Y. (2021). Inhibition of LDH-A by Oxamate Enhances the Efficacy of Anti-PD-1 Treatment in an NSCLC Humanized Mouse Model. Frontiers in oncology, 11, 632364. https://doi.org/10.3389/fonc.2021.632364

5. L355-371: This part should be placed under Institutional Review Statement or Ethical Approval. Funding information should be placed under ‘‘Funding’’ (combine it with L351-353). Please check the journal guidelines (https://www.mdpi.com/files/word-templates/ijms-template.dot):

Institutional Review Board Statement: In this section, you should add the Institutional Review Board Statement and approval number, if relevant to your study. You might choose to exclude this statement if the study did not require ethical approval. Please note that the Editorial Office might ask you for further information. Please add “The study was conducted in accordance with the Declaration of Helsinki, and approved by the Institutional Review Board (or Ethics Committee) of NAME OF INSTITUTE (protocol code XXX and date of approval).” for studies involving humans. OR “The animal study protocol was approved by the Institutional Review Board (or Ethics Committee) of NAME OF INSTITUTE (protocol code XXX and date of approval).” for studies involving animals. OR “Ethical review and approval were waived for this study due to REASON (please provide a detailed justification).” OR “Not applicable” for studies not involving humans or animals.

6. OTHER COMMENTS

In-text citations need to be aligned to the journal guidelines.

Other minor comments are highlighted in the manuscript PDF file attached.

Comments on the Quality of English Language

Moderate fixes are required

Author Response

Reviewer 3

The study entitled Characterization of a Humanized Mouse Model of Organophosphate Poisoning and Detection of Countermeasures via MALDI-MSI is a well-planned study with a reasonable contribution to the field. Humanization of Mouse Models in toxicological studies are not exactly new, nevertheless. I put forth the following suggestions.

  1. The abstract needs to be refined and reported in past tense, considering that this is a report.  

Response: The abstract as been updated to be entirely in the past tense.

  1. For keywords, words already used in the title should not be reused as keywords.

Response: We have updated the key words by removing those present in the title and adding new words.

  1. The methodology needs to make reference to previous studies from which these procedures were obtained from.

Response: The methodology was all developed for this publication.

  1. The discussion needs to be done in comparison with previous studies. In the current state, it can not be considered a discussion of the results. For example:

Qiao, T., Xiong, Y., Feng, Y., Guo, W., Zhou, Y., Zhao, J., Jiang, T., Shi, C., & Han, Y. (2021). Inhibition of LDH-A by Oxamate Enhances the Efficacy of Anti-PD-1 Treatment in an NSCLC Humanized Mouse Model. Frontiers in oncology, 11, 632364. https://doi.org/10.3389/fonc.2021.632364

Response: The discussion has been updated. Changes can be found via the tracked changes document.

  1. L355-371: This part should be placed under Institutional Review Statement or Ethical Approval. Funding information should be placed under ‘‘Funding’’ (combine it with L351-353). Please check the journal guidelines (https://www.mdpi.com/files/word-templates/ijms-template.dot):

Institutional Review Board Statement: In this section, you should add the Institutional Review Board Statement and approval number, if relevant to your study. You might choose to exclude this statement if the study did not require ethical approval. Please note that the Editorial Office might ask you for further information. Please add “The study was conducted in accordance with the Declaration of Helsinki, and approved by the Institutional Review Board (or Ethics Committee) of NAME OF INSTITUTE (protocol code XXX and date of approval).” for studies involving humans. OR “The animal study protocol was approved by the Institutional Review Board (or Ethics Committee) of NAME OF INSTITUTE (protocol code XXX and date of approval).” for studies involving animals. OR “Ethical review and approval were waived for this study due to REASON (please provide a detailed justification).” OR “Not applicable” for studies not involving humans or animals.

Response: The following statement has been added to the animal experiment section of the methods and we have updated the manuscript to contain a funding section:

The animal study protocol was approved by the Animal Care and Use Committee at the United States Army Medical Research Institute of Chemical Defense.

  1. OTHER COMMENTS

In-text citations need to be aligned to the journal guidelines.

Other minor comments are highlighted in the manuscript PDF file attached.

Response: These modifications have been made and can be seen via tracked changes in the manuscript.

Round 2

Reviewer 2 Report

Comments and Suggestions for Authors

the Authors have improved the paper significantly, therefore I suggest its publication in the present form

Reviewer 3 Report

Comments and Suggestions for Authors

All my comments have been satisfactorily addressed

Comments on the Quality of English Language

Minor fixes required